# Comparison of predictor approaches for longitudinal binary outcomes: application to anesthesiology data

Anil Aktas Samur[1], Nesil Coskunfirat[2] and Osman Saka[1]

[1] Faculty of Medicine, Department of Biostatistics and Medical Informatics, Akdeniz University, Antalya, Turkey
[2] Faculty of Medicine, Department of Anesthesiology and Reanimation, Akdeniz University, Antalya, Turkey

## ABSTRACT

Longitudinal data with binary repeated responses are now widespread among clinical studies and standard statistical analysis methods have become inadequate in the answering of clinical hypotheses. Instead of such conventional approaches, statisticians have started proposing better techniques, such as the Generalized Estimating Equations (GEE) approach and Generalized Linear Mixed Models (GLMM) technique. In this research, we undertook a comparative study of modeling binary repeated responses using an anesthesiology dataset which has 375 patient data with clinical variables. We modeled the relationship between hypotension and age, gender, surgical department, positions of patients during surgery, diastolic blood pressure, pulse, electrocardiography and doses of Marcain-heavy, chirocaine, fentanyl, and midazolam. Moreover, parameter estimates between the GEE and the GLMM were compared. The parameter estimates, except time-after, Marcain-Heavy, and Fentanyl from the GLMM, are larger than those from GEE. The standard errors from the GLMM are larger than those from GEE. GLMM appears to be more suitable approach than the GEE approach for the analysis hypotension during spinal anesthesia.

## INTRODUCTION

Longitudinal studies are designed to evaluate change within an individual over time. Repeated observations and covariates are conducted with these individuals. Because repeated measurements are made on the same subjects at different times, multiple assessments within subject responses are positively correlated. In analyzing longitudinal studies, this dependence must be accounted for in order to make correct inferences (*Fitzmaurice & Laird, 1993*; *Fitzmaurice, Laird & Rotnitzky, 1993*; *Laird & Ware, 1982*).

Several models have been proposed for the analysis of clustered data. A particular feature of longitudinal data is that they are clustered. The dependent variable is measured for each subject, and the subjects belong to a cluster, such as families, or classes. In longitudinal studies, the dependent variable is measured repeatedly for the same subject on different occasions, and subjects are clustered within the same unit. The dependent variables within the same cluster are assumed to be correlated (*Agresti, 2002*; *Fitzmaurice, Laird & Ware, 2004*).

Corresponding author
Anil Aktas Samur,
anilim@gmail.com

Most of these models are extensions of the generalized linear models with logistic, probit, or complementary log–log link functions (*Carriere & Bouyer, 2002*). These link functions are used for binary dependent variables. These models are usually classified into marginal or random effects models. Marginal models are also called population-averaged models, whereas random-effects models are also referred to as generalized linear mixed models (GLMM), or multilevel models.

In a marginal model, the entire response vector is modeled marginally on a set of covariates; the association structure is then typically captured via a set of association parameters, such as correlations, odds ratios, etc. The marginal model for the mean response depends only on the covariates of interest, not on any random effects or previous responses.

The generalized estimating equation (GEE) approach is the most popular method seen in marginal models. GEE is an extension of generalized linear models (GLM) for the analysis of longitudinal data. In this method, the correlation between measurements is modeled by assuming a working correlation matrix. This assumption eases the estimation of model parameters. Estimating the correct working correlation matrix provides efficiency parameter estimates. Even if it isn't correctly estimated, the model parameters from GEE tend to be consistent (*Hardin & Hilbe, 2007*).

Moreover, GLMM is an extension of GLM, inasmuch as it allows random effects in linear predictors. GLMM is useful for modeling the dependence among response variables in longitudinal or repeated measures studies, as well as for accommodating over-dispersion among responses. Over-dispersion refers to the presence of higher variability than expected in dataset. Over-dispersion may occur when assuming that a dependent variable has binomial distribution in GLMMs. This is because variance is a function of the mean for binomial distribution. It occurs when there is a correlation between observations, or observations are collected from clusters, or due to the heterogeneity of the subjects.

In GLMMs, the model is constructed with both a fixed and a random component. The fixed component usually estimates the experimental effect, whereas the random component estimates the heterogeneity across clusters in the regression coefficient (*Moscatelli, Mezzetti & Lacquaniti, 2012*).

The aim of this study was to compare the regression parameters and standard error of two analyses for longitudinal data. Additionally, in this paper we investigated the GEE and the GLMM approaches for predictor analysis in order to identify factors associated with hypotension during the intra-operative and post-operative period.

## GENERALIZED ESTIMATING EQUATIONS (GEE)

The generalized estimating equations (GEE) approach proposed by Liang and Zeger is, as previously noted, an extension of generalized linear models (GLM). GLM is a linear model and in the GLMs, the response variable has a distribution pattern seen in the exponential family. A GLM can be defined by three components. The first is the linear predictor $\eta$, which is a linear combination of regression coefficients:

$$\eta_i = x_i'\beta. \tag{1}$$

The second is the link function $g(.)$ that relates the mean of the data to the linear predictor:

$$g(E(Y_i)) = \eta_i. \tag{2}$$

The last component is the response distribution for $Y_i$ from the exponential family of distributions (*Agresti, 2002*; *Mccullagh & Nelder, 1989*).

The GEE approach is used for the analysis of correlated response data (*Dahmen & Ziegler, 2004*; *Liang & Zeger, 1986*; *Omar et al., 1999*). This method does not require distributional assumptions. GEE describes changes in the population mean and is used to estimate population average models or marginal models (*Fitzmaurice et al., 2008*). An advantage of this approach is that if the model for the mean has been correctly specified, consistent estimators can be obtained, even if other components of the model, such as the working correlation matrix, have been mis-specified (*Hardin & Hilbe, 2003*; *Warton, 2011*).

Let $y_i = (y_{i1}, \ldots, y_{ij}, \ldots, y_{in_i})^T$ represent the response vector for the $i$-th subject, where we assume that observations from the same subject are correlated or depend on each other to some extent. Observations from different subjects are assumed to be independent. The observed value $y_{ij}$ is related to the linear predictor $x_{ij}^T \beta$ towards the appropriate link function,

$$g(E(y_{ij})) = x_{ij}^T \beta \tag{3}$$

where $g$ is an appropriate link function, which identifies a function of the mean that is a linear prediction of covariates, e.g., identity for continuous response variables, or the logit function for binary response, and $\beta$ is a vector of regression coefficients. The variance is defined by

$$var(y_{ij}) = \phi V(E(y_{ij})) \tag{4}$$

where $V$ is a known variance function and $\phi$ is a possible unknown scale or over-dispersion parameter. The regression coefficient estimates, $\beta$ are defined by the solution of the GEE

$$\sum_{i=1}^{N} \frac{\partial \mu_i}{\partial \beta^T} V_i^{-1} (Y_i - \mu_i) = 0 \tag{5}$$

with $V_i = \emptyset A_i^{\frac{1}{2}} R_i(\alpha) A_i^{\frac{1}{2}}$, where $A_i$ is a $n_i \times n_i$ diagonal matrix with the variance of $Y_i$ as the $t$-th diagonal element and $R_i(\alpha)$ is the working correlation matrix of $Y_i$, indexed by a vector of parameters $\alpha$ (*Dahmen & Ziegler, 2004*; *Fitzmaurice et al., 2008*; *Kopcke et al., 2004*; *Liang & Zeger, 1986*; *Omar et al., 1999*).

## GENERALIZED LINEAR MIXED MODELS (GLMM)

The Generalized Linear Mixed Model (GLMM) is an extension of the GLM for clustered categorical data. The GLMM combines two statistical frameworks, which are the GLM and Linear Mixed models (LMM). GLMs combine regression models for different response types such as linear models for continuous responses, logistic models for binary responses, and log-linear models for counts. LMMs are linear regression models that include normally distributed random effects in addition fixed effects (*Fitzmaurice et al., 2008*).

In the LMM, it is assumed that the conditional distribution of each $Y_{ij}$, given a vector of random effects $b_i$, has a normal distribution, with $Var(Y_{ij}|b_i) = \sigma^2$. Furthermore, given the random effects $b_i$, it is assumed that the $Y_{ij}$ are independent of one another (given $b_i$, $Y_{ij}$ and $Y_{ik}$ are assumed to be independent of each other) (*Fitzmaurice, Laird & Ware, 2004*).

In the GLMM, it is assumed that the conditional distribution of each $Y_{ij}$, given a $q \times 1$ vector of random effects $b_i$, belongs to the exponential family of distributions (*Fitzmaurice, Laird & Ware, 2004*). The GLMM uses the inverse link function to describe the relationship between the linear predictor and the conditional mean (*Katrien & Jan, 2005*). The linear predictor for the GLMM:

$$g(E(Y_{ij}|b_i)) = \eta_i = X'_{ij}\beta + Z'_{ij}b_i \tag{6}$$

where $g(.)$ is a known link function. The variance of each of $Y_{ij}$, given a vector of random effects $b_i$,

$$Var(Y_{ij}|b_i) = v\{E(Y_{ij}|b_i)\}\phi \tag{7}$$

where $v(.)$ is a known variance function, a function of the conditional mean $E(Y_{ij}|b_i)$ (*Fitzmaurice, Laird & Ware, 2004*). Also, given the random effects $b_i$, it is assumed that the $Y_{ij}$ are independent of one another; this is the so called "conditional independence" assumption (*Fitzmaurice et al., 2008*). The random effects are assumed to have some probability distribution. Any multivariate distribution can be assumed for the $b_i$; in practice it is common to assume that $b_i$ have a multivariate normal distribution, with zero mean, and $q \times q$ covariance matrix, G. In addition, the random effects, $b_i$ are assumed to be independent of the covariates, $X_i$ (*Fitzmaurice, Laird & Ware, 2004*).

The GLMMs are the GLMs that include multivariate normal random effects in the linear predictor. Nevertheless, there is a difference between the GLM and the GLMM; this difference is error terms. The GLM with probit link function is:

$$\Phi^{-1}[P(Y_{ij} = 1)] = \beta_0 + \beta_1 x_{ij}. \tag{8}$$

Latent variable $Y^*_{ij}$ and the model is defined as;

$$Y^*_{ij} = \beta_0 + \beta_1 x_{ij} + v_{ij}. \tag{9}$$

The error term $v_{ij}$ is the sum of two error terms, such that:

$$\begin{aligned} v_{ij} &= u_i + \varepsilon_{ij} \\ u_i &\sim N(0, \sigma_u^2) \\ \varepsilon_{ij} &\sim N(0, \sigma_\varepsilon^2). \end{aligned} \tag{10}$$

The error term $\varepsilon_{ij}$ represents the variability within subjects and the other error term $u_i$ represents the variability between subjects. Additionally, the error term $u_i$ is also known as the random effects parameter (*Moscatelli, Mezzetti & Lacquaniti, 2012*).

In GLMMs the overall variability is separated into a fixed and a random component. The fixed component usually estimates the effect of interest, such as the experimental

effect, whereas the random component estimates the heterogeneity between clusters (i.e., between subjects) (*Moscatelli, Mezzetti & Lacquaniti, 2012*). The GLMM is used to analyze changes in individual response means, rather than population average. This model is therefore appropriate for modeling and for the prediction of individual response profiles.

## DESCRIPTION OF THE CLINICAL DATA

All of the cases that were admitted to the Akdeniz University Hospital Anesthesiology and Reanimation Department during the period of January 2008 to January 2011 were evaluated retrospectively. The records of 417 patients who had spinal anesthesia within this 3 year time period were obtained. Patients below 17 years old were excluded. 375 of those 417 patients were over 17 and were therefore included in the study.

Hypotension is common during spinal anesthesia (*Sharma, Gajraj & Sidawi, 1997*). According to the literature, hypotension has an incidence of 15%–33% (*Carpenter et al., 1992*; *Hartmann et al., 2002*; *Lin et al., 2008*). Certain studies have shown that people who receive anesthesia during the operation can die as a result of hypotension. According to studies defining the factors associated with hypotension, there are particular risk factors, such as age, gender, anesthesia drugs and doses (*Carpenter et al., 1992*; *Hartmann et al., 2002*; *Maxson, 1933*; *Tarkkila & Kaukinen, 1991*).

The outcome variable of interest in our study was hypotension. Interestingly, there is no universally accepted definition of hypotension in the literature. In a systematic review, *Klöhr et al. (2010)* highlighted the two most frequently used definitions of hypotension. Based on *Klöhr et al. (2010)*, we have therefore used the following definition, which is systolic blood pressure (SBP) <100 mmHg, or a decrease of <80% of the baseline SBP, to define hypotension.

$$Hypotension = \begin{cases} 1 \text{ (yes)}, & \text{if } SBP < 100 \text{ or } SBP < (baseline\ SBP) * 0.8 \\ 0 \text{ (no)}, & \text{if } SBP \geq 100 \text{ or } SBP > (baseline\ SBP) * 0.8 \end{cases}.$$

Our independent variables were:

  (i)  Patient's age (year);

  (ii)  Patient's gender: Male or Female

  (iii)  Surgical department: general surgery, urology, obstetrics and gynecology (O&G)

  (iv)  Positions of patients during surgery: Lithotomy, supine

  (v)  Diastolic blood pressure (DBP)

  (vi)  Pulse

  (vii)  Dose of Marcain-heavy

(viii)  Dose of Chirocaine

  (ix)  Dose of Fentanyl

  (x)  Dose of Midazolam

  (xi)  Electrocardiography (ECG).
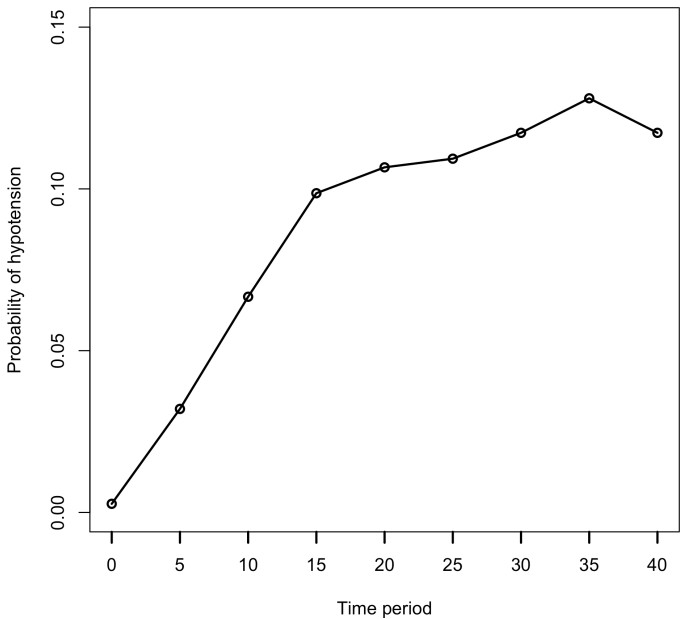

**Figure 1 Probability of hypotension by time period.** Each data point is shown as probability of hypotension in this graph. This plot reveals the variability in subject's hypotension level at entry, hypotension level at exit.

**Table 1 Descriptive statistics of dose of anesthetic drugs.**

| Anesthetic drugs | Min. | Median | IR | Mean | SD | Max. |
|---|---|---|---|---|---|---|
| Marcain-heavy | 0 | 9 | 12 | 7.11 | 6.18 | 25 |
| Chirocaine | 0 | 0 | 13 | 5.29 | 7.89 | 75 |
| Midazolam | 0 | 1 | 1 | 0.82 | 1.00 | 8 |
| Fentanyl | 0 | 0 | 0.05 | 0.03 | 0.06 | 0.20 |

**Notes.**
IR, Interquartile range.

There was no missing data in either the outcome or the covariates. The outcome variable was hypotension that had been diagnosed within a 40-minute period. Hypotension was recorded every 5 min during the surgery. Figure 1 shows the change in the probability of hypotension at each 5-min interval.

The mean (SD) age was 48.81 years (18.91) and the summary statistics for the doses of anesthetic drugs are given in Table 1.

In this study, 56% ($n = 210$) of patients were male, 44% ($n = 165$) were female. 38.4% ($n = 144$) of patients underwent surgery at the Department of obstetrics and gynecology, 44% ($n = 165$) at the urology service, and 17.6% ($n = 66$) underwent surgery at the general surgery service. In 41.9% ($n = 157$) of patients, surgery was performed in the lithotomy position, while 58.1% ($n = 218$) of patients were placed in the supine position. In 97.6% ($n = 366$) of patients, ECG was normal; in 2.4% ($n = 9$) of patients, it was abnormal. The summary statistics of DAP and Pulse are in Table 2.

**Table 2  Descriptive statistics of DAP and Pulse by time.**

| | DAP | | Pulse | |
| --- | --- | --- | --- | --- |
| | Mean ± SD | Median (IR) | Mean ± SD | Median (IR) |
| Baseline | 83.82 ± 14.06 | 80(20) | 87.54 ± 15.93 | 85(22) |
| 5 min | 78.58 ± 13.36 | 78(20) | 84.11 ± 14.92 | 80(20) |
| 10 min | 76.18 ± 13.56 | 75(20) | 82.46 ± 14.37 | 80(22) |
| 15 min | 75.65 ± 13.45 | 75(20) | 81.57 ± 14.28 | 80(20) |
| 20 min | 75.24 ± 12.98 | 75(20) | 80.94 ± 14.40 | 80(20) |
| 25 min | 75.08 ± 12.54 | 75(18) | 80.53 ± 14.40 | 80(20) |
| 30 min | 74.50 ± 12.21 | 75(15) | 79.99 ± 13.98 | 80(20) |
| 35 min | 74.19 ± 12.08 | 75(15) | 79.53 ± 13.76 | 80(20) |
| 40 min | 73.49 ± 12.12 | 75(15) | 78.94 ± 13.67 | 78(20) |

In Fig. 1, the probability of hypotension does not change regularly over time. There are different slopes. For this reason, it is assumed that every patient has a two-piece linear spline growth curve with a knot at the time of surgery. Piecewise regression was used to define a breakpoint for our study (*Bůržková & Lumley, 2007*; *Naumova, Must & Laird, 2001*). According to piecewise analysis, the summary of the breakpoint was found as 15.94 ± 1.18. In the current study, hypotension was measured at five-minute intervals, from beginning of the operation (0 min) to the 40th minute of each. We analyzed data using the GEE and GLMM at two time points (minutes 15 and 20 respectively) to decide the break points using the Akaike Information Criterion (AIC), which is a statistic of selecting model within a likelihood based model, and Quasi-likelihood under the independence model Criterion (QIC), which is a statistic for model selection for GEE models and analogous to AIC. Since the GEE method is a non-likelihood based, AIC is not used for GEE models.

When we compared the two models using the 20th minute time point and the 15th minute time point, better results were observed at 20 min. For the GEE model, QIC values were 1712.47 for 15 min and 1707.50 for the 20th minute. For the GLMM model, AIC values were 1137.56 for the 15th minute, and 1131.10 for the 20th minute. According to these two criteria, the model with the smaller statistic (AIC/QIC) is preferred. Therefore, we defined the cut off point for our study as the 20th minute. Figure 2A shows the estimated values of breakpoint according to piecewise regression analysis. Figure 2B shows the breakpoint as the 20th minute, and using the piecewise analysis, two straight lines for each patient were connected at the time of surgery.

## COMPARISON OF THE METHODS USING ANESTHESIOLOGY DATA

The marginal model was applied to the dataset with the GEE approach and the random effects model with GLMM approach. We used SAS (Version 9.2) procedures GENMOD with an independent working correlation matrix and GLIMMIX. In the GLMM, three random effects were determined: the random intercept, and two random slopes (time

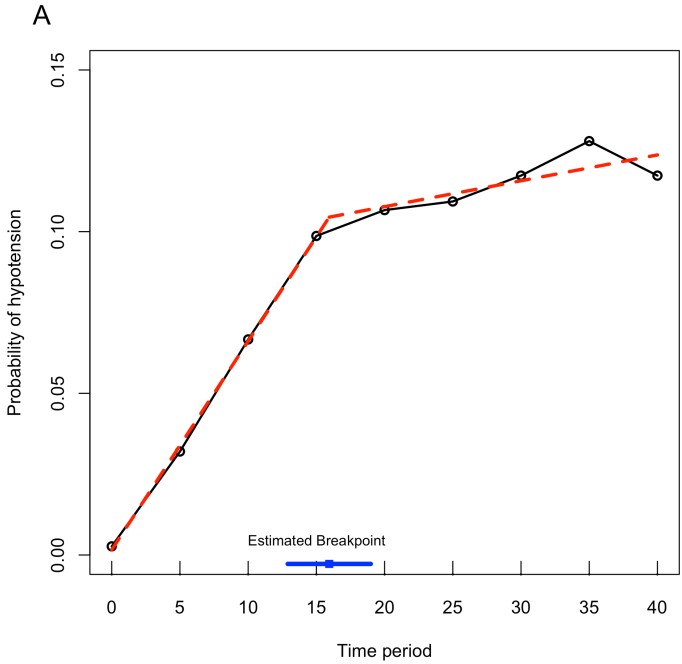

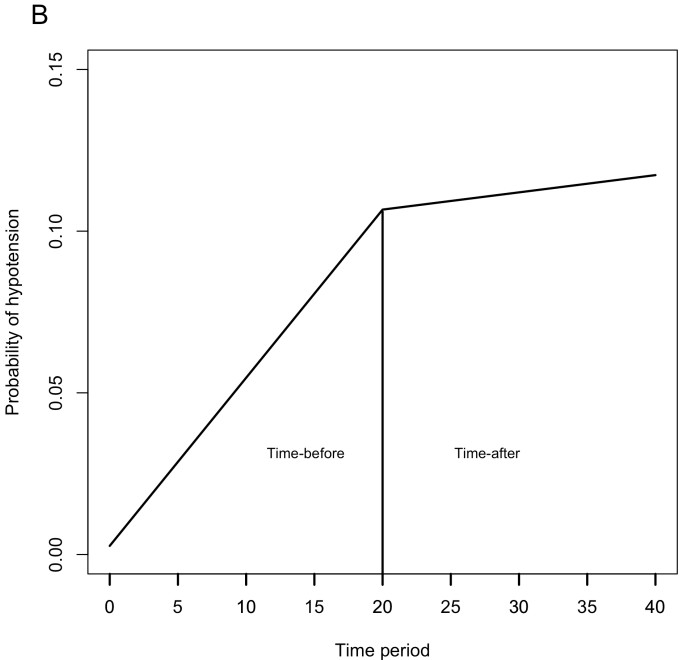

**Figure 2 The piecewise regression fit between time and probability of hypotension.** (A) This figure shows that the red trend line is calculated with piecewise linear regression analysis with breakpoint. The blue line shows the estimated breakpoint according to piecewise regression. (B) The breakpoint is defined as 20th minute. The line has an increasing trend also after the 20th minute.

**Table 3** **The results of marginal model and random-effects models for data.**

| | GEE | | | GLMM | | |
|---|---|---|---|---|---|---|
| | Estimate | SE | *p*-value | Estimate | SE | *p*-value |
| Intercept | 0.3949 | 1.8768 | 0.8333 | 0.6016 | 3.1718 | 0.8497 |
| Time-before | 0.0732 | 0.0116 | <.0001 | 0.1002 | 0.0253 | <.0001 |
| Time-after | 0.0428 | 0.0062 | <.0001 | 0.0323 | 0.0165 | **0.0500** |
| Age (year) | 0.0271 | 0.0091 | **0.0030** | 0.0469 | 0.0235 | **0.0463** |
| Gender (female) | 0.4146 | 0.5646 | 0.4628 | 1.4871 | 1.2435 | 0.2319 |
| Operation (urology) | 0.2770 | 0.4535 | 0.5413 | 1.4801 | 1.4207 | 0.2976 |
| Operation (O&G) | 0.6551 | 0.7203 | 0.3630 | 1.3287 | 1.0218 | 0.1936 |
| Position (supine) | 0.4013 | 0.3366 | 0.2331 | 0.7643 | 0.6850 | 0.2646 |
| ECG (normal) | −0.3070 | 0.7932 | 0.6988 | −0.5147 | 1.7085 | 0.7633 |
| DBP | −0.0863 | 0.0134 | <.0001 | −0.1941 | 0.0224 | <.0001 |
| Pulse | 0.0041 | 0.0096 | 0.6674 | 0.0244 | 0.0157 | 0.1204 |
| Marcain-heavy | −0.0028 | 0.0304 | 0.9264 | 0.0008 | 0.0702 | 0.9907 |
| Chirocaine | −0.0297 | 0.0238 | 0.2120 | −0.0488 | 0.0549 | 0.3734 |
| Fentanyl | 0.2940 | 2.1471 | 0.8911 | 0.2909 | 4.7399 | 0.9511 |
| Midazolam | 0.1199 | 0.0964 | 0.2137 | 0.3738 | 0.2474 | 0.1309 |

before and time after). For the two approaches, we tested the covariates and interaction terms. The interaction terms were removed from the models as they were non-significant. The results of comparison between the two models are shown in Table 3.

The results were similar for the 2 models. Although the parameters that include time-before, time-after, age, and DBP were common in both models, the parameter estimates were different. Differences between regression coefficients and between standard errors from marginal and random effects model are expected. The coefficients estimates from the GEE are lower in magnitude than corresponding coefficients estimates from the GLMM (*Fitzmaurice, Laird & Ware, 2011*) except time-after, Marcain-Heavy, Fentanyl. Furthermore, standard errors from the GLMM are larger than those from the GEE. The interpretation of coefficients of both models is different. When looking at Table 3, the estimated regression coefficient of the marginal model corresponding to time-before suggests that the log odds of a hypotension increase 0.0732 unit from the baseline to 20 min at every 5 min interval. The effect of gender (not significant) increased the logit of the probability of hypotension in the population of women more than in the population of men.

$$
\begin{aligned}
logit\{E(Y_{ij})\} = {} & \beta_1 + \beta_2 timebefore_{ij} + \beta_3 timeafter_{ij} + \beta_4 Age_i \\
& + \beta_5 Gender_i + \beta_6 Operation(U)_i + \beta_7 Operation(O\&G)_i \\
& + \beta_8 Position_i - \beta_9 ECG_i - \beta_{10} DBP_{ij} + \beta_{11} Pulse_{ij} \\
& - \beta_{12} MarcainHeavy_i - \beta_{13} Chirocaine_i + \beta_{14} Fentanyl_i \\
& + \beta_{15} Midazolam_i.
\end{aligned}
\tag{11}
$$

On the other hand, the result of the random effect model corresponding to time-before suggests that the log odds of probability of the hypotension for a patient increase 0.1002

units from the baseline to 20 min at every 5 min interval.

$$
\begin{aligned}
logit\{E(Y_{ij}|b_i)\} = {} & \beta_1 + \beta_2 timebefore_{ij} + \beta_3 timeafter_{ij} + \beta_4 Age_i \\
& + \beta_5 Gender_i + \beta_6 Operation(U)_i + \beta_7 Operation(O\&G)_i \\
& + \beta_8 Position_i - \beta_9 ECG_i - \beta_{10} DBP_{ij} + \beta_{11} Pulse_{ij} \\
& - \beta_{12} MarcainHeavy_i - \beta_{13} Chirocaine_i + \beta_{14} Fentanyl_i \\
& + \beta_{15} Midazolam_i + b_{1i} + b_{2i} timebefore_{ij} \\
& + b_{3i} timeafter_{ij}.
\end{aligned} \tag{12}
$$

Comparison of the two estimated coefficients for time-before, $e^{\hat{\beta_1}} = 1.08$ and $e^{\hat{\beta_1^*}} = 1.11$, respectively, from marginal and random effects models clearly show the distinction between these two methods.

Marginal models take into account the averaged relationship, but the random effects models express the relationships on inter-individual via random effects. In our study, although the results were similar, the estimates from the two models were different. The differences between parameter estimates of the two models largely depend on the between-individual heterogeneity. This heterogeneity can be described by random intercept and random slopes (time-before, and time-after) variances in the random model. The random intercept variance is 10.085, which is very high; this value indicates that there is great importance in between-patient variability in the propensity for hypotension, and it shows that within-subject association is strong. Approximately 95% of patients have a baseline risk of hypotension that varies from 0.03% to 99%. The random slopes variances are, respectively 0.003 and 0.007. Similarly, the 95% intervals of the random slopes variability vary respectively from 50% to 55% and from 47% to 55%. These values show the amount of variability in the slopes across patients. This inter-individual heterogeneity shows the differences between the parameters estimate of the marginal model and the random effects model.

## DISCUSSION

In longitudinal studies, repeated measures are correlated data that is taken from the same person at different times, and this correlation is important for analysis methods. There are various methods that have been proposed for the analysis of repeated binary data. The GEE, which is a marginal model, and the GLMM, which is a random effects model, are the two major and most common methods for analyzing such cases.

In general, parameter estimates and standard errors from random effect models are greater than those from marginal models. The difference in the estimates between these two models is due to the correlation between repeated measures. The interpretation of covariates in random effects models is more difficult than in marginal models. For random effect models, the interpretation of estimates is related to changes within subjects. However, marginal models ignore such changes within subjects. This is due to the fact that the target of marginal models is the population, while the target of random effect models is the subject.

In marginal models, the regression coefficient describes how the average rates for any variable may be changed in the study population. The exponential of an estimate parameter represents a population-averaged odds ratio for the response and relates to the sub-population that includes the covariate concerning the sub-population not including the covariate. In the random models, the regression coefficient describes how the odds of any variable for any patient are subject to change. The exponential of the estimate parameter is an odds ratio for a person that has a covariate, when compared to the same person not having a covariate (*Fitzmaurice, Laird & Ware, 2004*; *Hubbard et al., 2010*).

The strength of this study lies in the longitudinal nature of the data set. Nevertheless, there were certain limitations to analyzing both methods. The GEE method is not difficult to apply and is now available in the major statistical analysis packages, but the procedures are more complex for the GLMM. What's more, the GEE model does not allow for assessing the suitability of fit (*Odueyungbo et al., 2008*), whereas the GLMM does (*Moscatelli, Mezzetti & Lacquaniti, 2012*).

Marginal models are popular for binary longitudinal data. However, the choice of method, GEE or GLMM, depends on the aim of the particular study. Marginal models are appropriate if the research focus is on population-average, but if it focuses on individual differences, random effect models are appropriate.

In GLMMs, the fixed effects parameters $\beta$ have conditional interpretations, given the random effects. There are two types of fixed effects. The first of these is that the effect of an explanatory variable refers to the effect on the response of a within-cluster, or within-subject (i.e., subject-specific) 1-unit increase of that predictor. The other is that the effect of an explanatory variable is a between-cluster. It is in this sense that random effects models are conditional models, as both within- and between-cluster effects apply conditionally on the random effect value (*Agresti, 2002*). On the other hand, effects in marginal models refer to overall clusters (i.e., population-averaged). For the logit model, the difference between the two models is that the population-averaged effects are smaller than the cluster-specific effects (*Agresti, 2002*). There are approximate relationships between estimates from these two models in the logistic-normal case. The effect in the marginal model multiplies that of the conditional model by about c (*Zeger, Liang & Albert, 1988*); it is typically smaller in absolute value. The discrepancy increases as $\sigma$ increases (*Agresti, 2002*).

Figure 3 shows that when the marginal effect was compared to the subject-specific effect, the marginal effect is smaller than the random effects. For a single explanatory variable and some subjects, the figure shows subject specific curves for $P(Y_{it} = 1|b_i)$ when considerable heterogeneity exists. This corresponds to a relatively large $\sigma$ for the random effects. At any fixed value of the explanatory variable, variability occurs in the conditional means $E(Y_{it}|b_i) = P(Y_{it} = 1|b_i)$. The average of these is the marginal mean, $E(Y_{it})$ (*Agresti, 2002*). Focusing on the range of probability of hypotension between 0.2 and 0.8, the population-averaged effect, that is the logistic curve, is more linear, but the slope curves of subject-specific curves rise more rapidly than the marginal slope of probabilities.

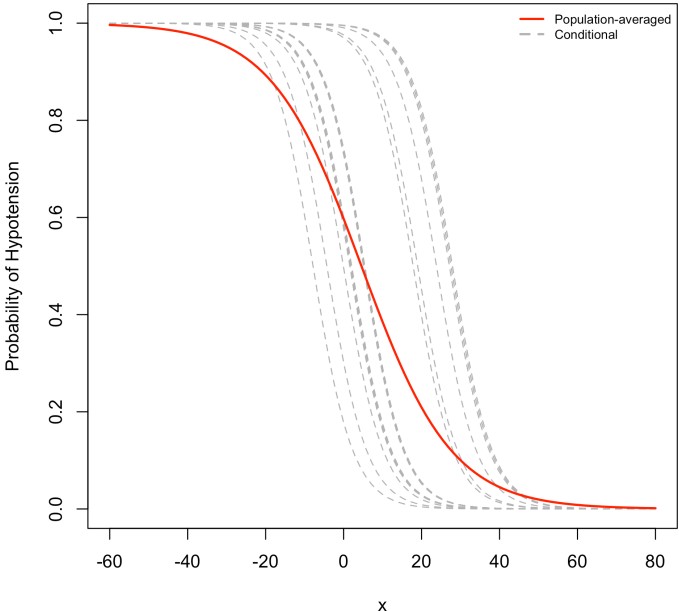

**Figure 3 Comparison of random effects model and marginal model.** In this figure, the conditional of probabilities of hypotension (dotted lines) and marginal probability of hypotension (solid line) are compared for a single explanatory variable, and for several subjects.

## CONCLUSION

In clinical researches, longitudinal studies with binary repeated events are frequently undertaken. Nevertheless, traditional analyses are inefficient for such studies, and the selection of a more efficient model, namely marginal models or random effects models, has been the primary focus of this study. As is shown, marginal models and random effect models are useful for longitudinal data.

In this study, we compared both methods and found that the regression parameters from GEE are smaller than those from GLMM, while all except three variables and all standard errors from GEE are smaller than those from GLMM.

The individual characteristics of each patient given spinal anesthesia are valuable in terms of understanding the change of probability of hypotension. The identification factors associated with hypotension during anesthesia, such as the type and position of surgery, and the anesthesia drugs and doses, change vary according to the individual differences of patients. The marginal model, GEE, does not measure the association between the change within-subject covariate and the change in the outcome. For this reason, GLMM appears to be more suitable for the analysis of hypotension during spinal anesthesia.

## ACKNOWLEDGEMENTS

The authors are most grateful to Prof. Nan Laird for her valuable suggestions regarding statistical analysis. We also thank Dr. Hanife K. Kabukcu for her assistance with data collection.

### Funding

The authors declare there was no funding for this work.

### Competing Interests

The authors declare there are no competing interests.

### Author Contributions

- Anil Aktas Samur conceived and designed the experiments, performed the experiments, analyzed the data, contributed reagents/materials/analysis tools, wrote the paper, prepared figures and/or tables, reviewed drafts of the paper.
- Nesil Coskunfirat performed the experiments, contributed reagents/materials/analysis tools, reviewed drafts of the paper.
- Osman Saka conceived and designed the experiments.

### Human Ethics

The following information was supplied relating to ethical approvals (i.e., approving body and any reference numbers):

Ethical approval for the study was obtained from Scientific Research Projects Coordination Unit of Akdeniz University (approval number 01.03.2011/50).

### Supplemental Information

Supplemental information for this article can be found online at http://dx.doi.org/10.7717/peerj.648#supplemental-information.

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
