# Peer review of "Comparison of predictor approaches for longitudinal binary outcomes: application to anesthesiology data"

_PeerJ, doi:10.7717/peerj.648_

## Round 0.1 · original submission · Major Revisions

I concur with the reviewers that the manuscript is not acceptable in its current form. The aim of the paper is a comparison of two analytical approaches to the same data set, but the comparison of the two methods is not very clear and the basis for the authors' conclusion should be presented in more detail.

The methods used should be described in more detail, the figures are not publication quality and the text would benefit from a revision of the language.

·

Basic reporting

The reporting is clearly done, except the quality of the figures that should be improved from an edition point of view: A more detailed description of both figures (notably with regards to the data from which they have been plotted) should be provided as well as some mesure of precision in plotted estimates.
Some abbreviations appear undefined (for instance: SBP, for systolic blood pressure; IR, I guess for interquartile range, etc.).

Experimental design

This papers aims at comparing two modeling strategies of longitudinal binary data, namely generalized estimating equations (GEE) approach that deals with marginal effects, and generalized linear mixed models (GLMM) that are random effects models that allow estimating individual responses rather than population average. It uses a clinical dataset of 375 observations recorded to assess the predictive factors of hypotension after anesthesia, within (and not at) 40 min, from repeated measures every 5 minutes on all these patients.

Thus, the paper is is an application of two modeling strategies on a clinical data example.

Validity of the findings

The models used to fit the clinical data should be more clearly stated notably with regards of the time variables (the equations should be given). Moreover, interaction terms appear to have been ignored, while I guess there are the quantities of interest from the prognostic point of view.
The definition of the threshold should be discussed, and sensitivity analyses performed to assess its impact on results.

Additional comments

The paper should be improved with regards to the application to the clinical dataset. It should improve the presentation of the models (notably with respect to the introduction of the time variables in the models), to provide a more direct comprehension of the reported estimates. Moreover, given that the time course of hypotension appears to be different, interaction terms should be incorporated, in order to assess the impact of patient characteristics on such a time pattern of hypotension.

The figures should be improved to better display the precision in estimates.

Reviewer 2 ·

Basic reporting

The authors compare two statistical models, the GLMM and the GEE, for the analysis of longitudinal clinical data. The two models identified different factors predicting an episode of hypotension during the intra- and post-operative period. The paper is clearly written; yet I would recommend a reading proof by a native English speaker for language issues. The introduction covers the relevant background.

The authors often use technical terms without sufficient clarification. For example, on line 42 the authors introduce the term “clustered data”; I would recommend to briefly explain why this apply to longitudinal studies (that is, each subject is a cluster). Also, on line 58: The authors should explain “overdispersion” in models assuming a binomial response variable. The same apply in the next paragraph with the term “link function”.

Line 43 “logistic link function”: The authors might briefly mention other link functions such as the probit and the complementary log-log.

I found that the description of the GLMM was quite confusing. The authors should definitely improve this part. See below for some suggestions:

Lines 79-80, rephrase: “The observed values y_ij is related to the linear predictor X_ij^T Beta towards the appropriate link function”

Lines 91-93: Simply state that: “GLMM is an extension of GLM to clustered categorical data”.

Line 94: Why nonlinear mixed models are mentioned here?

The model equation on lines 97-98 is a LMM (identity link function). Why is it there?

The authors do not state in the text that b_i and epsilon are the two error terms – this is crucial to understand the model!

In the definition of Hypotension, define SBP (systolic blood pressure?).

Line 205, “The GEE does not allow assessing the goodness of fit”: Add citation.

Experimental design

I can’t find in the manuscript the model specification for the GLMM: How many random predictors did the authors specified in the model? Did the authors evaluate different GLMM specifications?

Both GEE and GLMM assumes that the effect are linear in the logit scale. Why the authors expect the relationship to be linear on the response (probability) scale (Figure 1-2)? Shouldn’t the response be linearized with respect to the logit of the probability?

Validity of the findings

The authors report that the parameters estimates from GLMM are larger than those from GEE. Actually, this larger estimate in a conditional model (GLMM) compared with the marginal model (GEE) is well established in the literature. For example in Agresti, Categorical Data Analysis, the difference between conditional and marginal model is discussed in paragraph 12.2.2 and clearly illustrated in figure 12.1. From the Agresti’s book:

“In fact, a fundamental difference between the two model types is that when the link function is nonlinear, such as the logit, the population-averaged effects of marginal models often are smaller than the cluster-specific effects of GLMMs.”

In the logistic-normal case, the effect in the marginal model multiplies
that of the conditional model by a factor c, which depends on the variance of the random effect (Zeger et al., 1988). The authors should refer to these relevant references in the discussion.

I would also suggest for, a selected predictor, to generate a figure comparing the subject-specific (conditional) effect (~10-20 representative subjects) and the marginal effect, on the same vein of figure 12.1 in the Agresti’s book. This would help clarifying – for the reader with basic statistical background – the difference between conditional and marginal models.

The final conclusion that GLMM is more suitable for the analysis of this type of data does not clearly emerge from the results.

Additional comments

A caveat: I am quite familiar with GLMM, whereas I have only basic knowledge about GEE. I am not able to tell if the GEE equations reported in the manuscript are correct (also: equation numbers are missing!).

Reviewer 3 ·

Basic reporting

This manuscript adresses the comparison of two major statistical methods those can be used for the analysis of longitudinal data with repeated binary outcomes. Study uses a well defined anesthesiology data set and aims to determine the performances of the methods for estimation of hypotension using the available demographical and clinical data.
Below, several important considerations are provided for the authors in order to increase the overall quality of the article.
1- The "Ethics Statement" is better to be placed at the end of the article.
2- I would recommend a review of the English, since it is obvious that none of the authors are native speakers. Several refinements are required to increase the readibility and eliminate grammatical errors.
3- Both models identify "age" as an important factor to influence hypotension risk. Did authors consider categorizing the continuous "age" (into two or more equal age intervals) and investigate its affect among different age categories?
4- Table 1 demonstrates the anesthetic drugs those were used during the surgeries, but there is no clear evidence in the article about the usage of these drugs. Authors should clearly explain the differences in doses and combinations of these drugs, since the variations of drugs used are unclear to the readers. Also the drug combinations might have different affects and such affects seem to be not considered during the whole analysis.
5- As shown in Figure 5, the cut off time set by the authors is shown to be 20 minutes. I would recommend the authors to clearly explain why this cut off time was set as 20 minutes rather than any other value (such as 15 or 30 minutes).
6- Discussion section is lacking the actual "discussion" of the study findings and it is constructed in a way very similar to a standard "introduction" section. Authors should expand this part and add some actual discussion content rather than introduction-like theoratical information.

Experimental design

No comment

Validity of the findings

The findings are all related to the dataset which was under consideration for the study. Thus, the clinical findings are only valid for the specified dataset. When multiple statistical analysis methods are being compared based on their appropriateness and performance, it would be beneficial to use multiple datasets and compare the the methods accordingly. Since the study focuses only on a single dataset, authors should express the fact that their clinical outcomes are directly related to the quality of the data and might not be generalized as a definite final outcome. It would be beneficial to underline this important issue, especially in the "Discussion" section.

---

## Round 0.2 · accepted · Accept

Overall clarity of the presentation has improved and I find the manuscript suited for publication in its present form.